# Capturing Temporal Dynamics in Large-Scale Canopy Tree Height Estimation

**Jan Pauls** [1] [*]  **Max Zimmer** [2] [*]  **Berkant Turan** [2] [*]
**Sassan Saatchi** [3]  **Philippe Ciais** [4]  **Sebastian Pokutta** [2]  **Fabian Gieseke** [1] [5]

## Abstract

With the rise in global greenhouse gas emissions, accurate large-scale tree canopy height maps are essential for understanding forest structure, estimating above-ground biomass, and monitoring ecological disruptions. To this end, we present a novel approach to generate large-scale, high-resolution canopy height maps over time. Our model accurately predicts canopy height over multiple years given Sentinel-1 composite and Sentinel 2 time series satellite data. Using GEDI LiDAR data as the ground truth for training the model, we present the first 10 m resolution temporal canopy height map of the European continent for the period 2019–2022. As part of this product, we also offer a detailed canopy height map for 2020, providing more precise estimates than previous studies. Our pipeline and the resulting temporal height map are publicly available, enabling comprehensive large-scale monitoring of forests and, hence, facilitating future research and ecological analyses.

## 1. Introduction

As global carbon emissions continue to rise, meeting the goals of the Paris Agreement[1] requires a comprehensive understanding of all climate-related factors. This includes precise quantification and temporal monitoring of carbon sinks. However, despite decades of research and the development

[*]Equal contribution [1]Department of Information Systems, University of Münster, Germany [2]Department for AI in Society, Science, and Technology, Zuse Institute Berlin, Germany [3]Jet Propulsion Laboratory (JPL), California Institute of Technology, USA [4]Laboratoire des Sciences du Climat et de l'Environnement, LSCE/IPSL, France [5]Department of Computer Science, University of Copenhagen, Denmark. Correspondence to: Jan Pauls <jan.pauls@uni-muenster.de>.

*Proceedings of the $42^{nd}$ International Conference on Machine Learning*, Vancouver, Canada. PMLR 267, 2025. Copyright 2025 by the author(s).

[1]https://unfccc.int/process-and-meetings/the-paris-agreement

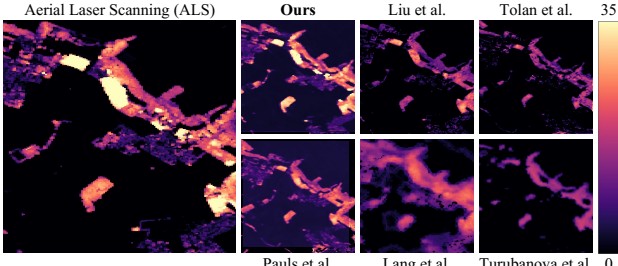

*Figure 1.* Comparison of six canopy height maps with precise measurements obtained via aerial laser scanning (ALS). The patches contain tall trees exceeding 30 m in height. Our model is the only one that can accurately estimate the height of such trees.

of numerous methods, this quantification remains insufficient for detailed and effective policymaking (Cook-Patton et al., 2020; Cuni-Sanchez et al., 2021). An essential part of this quantification is the monitoring of forest ecosystems, in turn allowing their management and conservation. To support climate adaptation and mitigation strategies, accurate and up-to-date information on forest health and carbon balance is critical to evaluate the current state of forests, to implement measures to prevent forest loss, and to improve management strategies (Friedlingstein et al., 2019).

A key approach to assess forest conditions is based on measuring or estimating tree heights, which results in height values that are used to approximate wood volume, so-called above-ground biomass and, consequently, the carbon stored in trees (Schwartz et al., 2023). A traditional way to obtain corresponding measurements is to manually measure individual trees, resulting in so-called National Forest Inventories (NFI). Such inventories are essential for forest monitoring, but are very costly and also lack global reach. This issue is worsened by varying forest monitoring efforts and techniques across different nations with different financial resources (Sloan & Sayer, 2015). However, advances in Earth observation and machine learning now enable automated, comprehensive global forest assessments by leveraging satellite data, including optical, radar, and LiDAR measurements (Hu et al., 2020). The resulting high-resolution canopy height maps are essential for understanding forest dynamics and supporting climate change mitigation efforts.

Recent studies have advanced canopy height prediction using classical machine learning methods (Potapov et al.,

2021; Kacic et al., 2023; Loo & Wang, 2024) and deep learning approaches (Schwartz et al., 2024; Fayad et al., 2023; Lang et al., 2023; Pauls et al., 2024; Liu et al., 2023; Tolan et al., 2024; Fayad et al., 2025), with the satellite-dependent resolution varying between 30 m (Landsat), 10 m (Sentinel-1, -2), 3 m (PlanetLabs) and 60 cm (Maxar) with only the Sentinel and Landsat program being openly available. However, almost all studies focus on predicting canopy height for a single year, despite that being insufficient (Bevacqua et al., 2024). Temporal dynamics of forests are essential for identifying carbon sources and sinks and assessing forest responses to natural or human-induced stressors such as drought, diseases, or environmental changes. Although some methods have explored temporal tree height mapping[2] at local scales (Kacic et al., 2023) and Turubanova et al. (2023) produced a map for Europe, no approach has yet tackled this challenge at a resolution of 10 m and greater scale combined, which is vital for assessing forest details.

In this work, we introduce an approach for generating large-scale, temporal tree height maps. Specifically, we use a custom deep learning approach that predicts tree canopy height from Sentinel image data, using the sparsely-distributed space-borne GEDI LiDAR data as a ground truth.

**Contributions.** We address the task of estimating tree canopy heights given satellite time series data as input. The main contributions made in this work are as follows:

1. We provide a model capable of accurately tracking forest height changes at 10 m resolution across the entire European continent, enabling consistent detection of growth and decline from 2019–2022.

2. We present a canopy height map of Europe for the year 2020, providing more accurate measurements and finer spatial details than previous studies, both in quantitative and qualitative assessments.

3. We demonstrate that using a 12-month time series of Sentinel-2 imagery, rather than using a single aggregated composite, yields substantial performance gains.

The entire pipeline and model weights are publicly released on GitHub[3] and the resulting tree canopy height maps are accessible through Google's Earth Engine[4] (Gorelick et al., 2017), ensuring reproducibility and facilitating research on large-scale forest monitoring, forest structure analysis and above-ground biomass estimation.

---

[2]We define *temporal* maps as those which provide canopy heights over multiple years, independent of the generation process.

[3]https://github.com/AI4Forest/
Europe-Temporal-Canopy-Height

[4]https://europetreemap.projects.
earthengine.app/view/europeheight

## 2. Background

Tree canopy height estimation has advanced significantly through integrating satellite data from Sentinel (European Space Agency, 2024), Landsat (Williams et al., 2006), GEDI (Dubayah et al., 2020), ICESat (Abdalati et al., 2010), and Aerial Laser Scanning (ALS). The goal is to predict the tree height for each pixel in satellite images. However, ground truth data like GEDI measurements are limited and sparsely distributed across the globe. This section covers the fundamental challenges of creating such maps.

### 2.1. Satellite Imagery

Openly available satellite imagery for tree canopy height prediction is primarily sourced from three key missions: Landsat, Sentinel-1, and Sentinel-2. The Landsat program (Williams et al., 2006), run by NASA since 1972, offers optical multi-spectral imagery with a 30 m resolution and a 16-day revisit cycle, meaning that new data are collected for any region every 16 days. The Sentinel missions, operated by the European Space Agency (ESA) since 2014, include Sentinel-1 with Synthetic Aperture Radar (SAR) and Sentinel-2 with multispectral sensors. Both provide 10 m resolution images and a global revisit time of about 5 days, allowing more frequent forest monitoring. While airborne data can achieve higher resolutions of up to 10 cm, it is often limited to specific regions and not widely accessible, making spaceborne imagery the preferred choice for large-scale height mapping.

While these satellite missions provide global-scale image data, single images are often not suitable for canopy height prediction. Radar satellites like Sentinel-1 can suffer from rain interference and noise, while multispectral sensors like Sentinel-2 and Landsat struggle with cloud and cirrus penetration. These images require preprocessing, such as cloud removal, color correction, and atmospheric correction, to convert from Top-of-the-Atmosphere (TOA) to Bottom-of-the-Atmosphere (BOA) images.

To further address these issues, temporal composites are used to aggregate images over time, employing techniques like per-pixel median calculation (Pauls et al., 2024; Schwartz et al., 2024; Fayad et al., 2023) or the Best-Available-Pixel approach (Senf & Seidl, 2021). These methods mitigate adverse weather and atmospheric effects, ensuring more consistent and reliable images for tree canopy height analysis. However, temporal aggregation often results in significant information loss, discarding valuable seasonal variations such as leaf-on vs. leaf-off seasons. This hinders capturing crucial temporal dynamics, but has thus far only been tackled for crop yield prediction and classification (Fan et al., 2021; Johnson & Mueller, 2021) and general satellite image processing (Tarasiou et al., 2023).

Additionally, aggregation removes the spatial shifts caused by geolocation inaccuracies, thus losing the opportunity to leverage this variability for more accurate predictions. Precisely, Sentinel-2 imagery exhibits a mean geolocation offset of about 4 m (Yan et al., 2018), causing each pixel's reflectance to be influenced by its surroundings. Leveraging this offset across multiple images can enhance the model's ability to detect structural details, such as distinguishing a large tree's borders within a forest of smaller trees. This results in more precise height predictions compared to using aggregated images. Wolters et al. (2023) demonstrated that using multiple Sentinel-2 images outperforms single-image methods by leveraging temporal information and geolocation shifts. Our work further builds upon this approach.

## 2.2. Tree Canopy Height Estimation

Tree canopy height mapping has evolved from classical machine learning (Kacic et al., 2023; Potapov et al., 2021) to advanced methods such as convolutional networks (Liu et al., 2023; Yan et al., 2018; Lang et al., 2023) and vision transformers (Fayad et al., 2023; Tolan et al., 2024).

A key challenge in training models for temporal canopy height estimation is obtaining accurate ground-truth data. This data can come from national forest inventories or LiDAR measurements, either airborne or spaceborne. Airborne LiDAR provides high-resolution data but is limited to local areas. In contrast, NASA's GEDI mission offers broader geographic and temporal coverage with spaceborne LiDAR, though at a coarser spatial resolution and with occasional geolocation inaccuracies. GEDI measures canopy heights[5] using laser shots with a 25 m footprint, but only about 4% of Earth's surface is covered during the satellite's operational period, and spatial alignment across different years is virtually non-existent. This inconsistency complicates the use of GEDI as a reliable temporal training dataset.

In consequence, most studies focus on estimating height for a single year, with few addressing multi-year time series (Dixon et al., 2025; Kacic et al., 2023; Turubanova et al., 2023). Mapping efforts over multiple years or seasons face significant computational and storage challenges and can result in fluctuating height estimates due to varying satellite conditions and model uncertainties. A common approach is to just apply a single-year model to data from other years, but this fails to capture temporal patterns. Consequently, post-processing techniques like moving averages, trend detection, and cut-detection are used to smooth out year-to-year variations. Developing a robust, large-scale framework for temporal tree canopy height mapping remains one of the key research challenges in forest monitoring.

## 3. Approach

Our methodology integrates multi-source satellite imagery and GEDI LiDAR data to produce high-resolution, temporal canopy height maps for Europe. Next, we detail the data sources, preprocessing steps, and the design choices made.

### 3.1. Data

To construct our dataset, we integrate information from three key sources: Sentinel-1, Sentinel-2, and GEDI. Each dataset undergoes rigorous preprocessing to ensure compatibility and optimal quality. To keep the best projection accuracy, we use a tiling system following the Universal Transverse Mercator (UTM)[6] system and process the data in these tiles.

**Sentinel-1 and Sentinel-2.** Sentinel-1 provides Synthetic Aperture Radar (SAR) measurements in two polarizations: VV (vertical-vertical) and VH (vertical-horizontal), acquired in the Interferometric Wide Swath (IW) mode. Images are collected from both ascending and descending orbits, resulting in four distinct channels per tile. Due to the high noise levels inherent in SAR data, we aggregate the measurements over temporal space by computing the per-pixel median across all acquisitions within a year, thereby mitigating noise and enhancing temporal consistency. The data is collected in the Sentinel UTM coordinate system, where each tile spans $100\,\text{km} \times 100\,\text{km}$.

Sentinel-2 multispectral imagery is a key part of our approach. We use the Level-2A surface reflectance product (BOA) available via the Copernicus AWS, selecting a single best image each month based on minimal cloud cover, thereby ensuring minimal contamination and consistent data quality. This yields a temporal sequence of 12 monthly images per tile and year (one per month), thereby preserving seasonal patterns important for vegetation monitoring. We include all Sentinel-2 bands except B10 (cirrus). Each band's reflectance values are normalized by a fixed factor (cf. Table 1), mapping values to $[0, 1]$. Rather than employing conventional min-max or zero-mean normalization techniques, which are substantially affected by the presence of clouds and cirrus, we identified value ranges for each band that contain "valuable" information, ensuring data quality is maintained for analysis. By retaining monthly variability instead of aggregating temporal data, our approach captures seasonal dynamics, such as transitions between leaf-on and leaf-off states, which are crucial for vegetation monitoring.

**GEDI.** GEDI LiDAR data provides sparse height measurements that are essential for model supervision. We use

---

[5]Although GEDI not only measures trees, this can be neglected here, as forest masks are used for filtering before further analysis.

[6]The UTM coordinate system divides the Earth into 120 zones, each $6°$ of longitude wide, using transverse Mercator projections to minimize distortion. Each zone is further divided into tiles following the Sentinel-2 tiling system for better data handling.

*Table 1.* Grouped band normalization for Sentinel-2.

| Bands | Divisor |
|---|---|
| B1 (Coastal) | $0.9 \times 10^3$ |
| B2, B3, B4 (Visible), B5 | $1.8 \times 10^3$ |
| B6, B7 (Red Edge), B11, B12 (SWIR) | $3.6 \times 10^3$ |
| B8, B8A (NIR), B9 (Water Vapor) | $5.4 \times 10^3$ |

the Level-2A product, transforming its geolocations to align with the Sentinel UTM grid. Several filters are applied to ensure data quality: valid quality flags, non-degraded shots, and a sensitivity threshold of $0.9$. We consider only high-power beams (IDs 5–8) due to their superior signal-to-noise ratio. Height values are constrained to a plausible range of $[0, 100\,\mathrm{m}]$ to exclude outliers. We utilize the `rh_98` metric (relative height 98%), which corresponds to the height below which 98% of returned photons are recorded, providing a robust estimate of canopy height. We use data from the years 2019–2022, where we have full coverage.

### 3.2. Model Architecture

We adopt a three-dimensional (3D) U-Net inspired by Çiçek et al. (2016), originally proposed for volumetric segmentation tasks, and modify it to output a single-channel tree canopy height for each input pixel. Unlike prior approaches relying on static images or single composites (Schwartz et al., 2024; Pauls et al., 2024), our model processes a stack of 12 monthly Sentinel-2 images concatenated with an aggregated Sentinel-1 composite. This approach allows the model to use potential seasonal vegetation changes, and possibly exploit geolocation offset in Sentinel-2 imagery by incorporating information from neighboring pixels.

The architecture follows the standard encoder–decoder U-Net structure, employing 3D convolutions throughout to capture both spatial and temporal dependencies. We adapt the final output layer to generate a single-channel height map rather than a multi-class segmentation mask. In parallel, Sentinel-1 data – aggregated into a single median-based image over the year – is repeated across these time slices so that the model can exploit both radar and optical signals. The encoder path slowly reduces the temporal dimension from its initial size down to 1 at the bottleneck, and each skip connection likewise applies a 3D convolution with a kernel size matching the remaining temporal images, effectively collapsing the temporal dimension to a single slice.

### 3.3. Model Training

Our training strategy is designed to address the challenges of sparse ground truth data, geolocation inaccuracies, and the need for temporally consistent predictions. The final model, based on the 3D U-Net architecture, is trained to

predict canopy heights across spatial and temporal scales using a modified Huber loss (Pauls et al., 2024).

**Modified Huber Loss.** Our final loss function is the Huber loss, as it effectively balances penalizing small errors while being less sensitive to outliers compared to L2 loss. This is especially important because GEDI geolocation data often contains outliers. Since GEDI's geolocation is not perfectly precise, we incorporate a shift mechanism (Pauls et al., 2024), to account for systematic geolocation offsets. This mechanism allows the model to adjust an entire flight path (referred to as a "track") by a predefined maximum offset, such as $10\,\mathrm{m}$, to reduce errors caused by consistent misalignment. Additionally, because the labels in our dataset are sparse – meaning that many pixels lack corresponding label values – we compute the loss only for labeled pixels and ignore unlabeled pixels during training.

**Optimization Setup.** We use Adam (Kingma, 2014) with an initial learning rate of 0.001, weight decay of 0.01, and gradient clipping at 1.0. We follow best practices (Li et al., 2020; Zimmer et al., 2023) and use a linear learning rate scheduler with a 10% warmup, training for $400{,}000$ iterations with a batch size of 16 (corresponding to 8 epochs).

**Dataset Size.** Our dataset comprises $800{,}000$ randomly selected patches, each measuring $2.56\,\mathrm{km} \times 2.56\,\mathrm{km}$ (approx. $0.15\%$ pixels have labels per patch), totaling $8\,\mathrm{TB}$ in size. To minimize computational load and data transfer, we use a 10% subset for training and hyperparameter tuning for the different baselines, as outlined in Section 4.1. The final model is then trained on the entire dataset.

**Post-Processing.** A problem of temporal height maps is fluctuating predictions due to uncertainties in the data, such as inconsistent color calibration and varying cloud cover. To mitigate this, we apply a quadratic smoothing spline to capture the underlying trend while reducing noise. We set the smoothing parameter to 5 to balance fidelity to the data with improved temporal consistency, ensuring a clear and more interpretable prediction visualization.

## 4. Results

This section presents the results of our temporal tree canopy height model. We focus on the accuracy of our predicted tree canopy height maps, comparisons with existing models, and generalization across european forest regions.

All results in this section are based on 1,500 randomly selected validation points, see Figure 2 for the distribution. At each validation point, we select a $2.56\,\mathrm{km} \times 2.56\,\mathrm{km}$ area to collect all available GEDI labels that match the same criteria as in Section 3.1. We compare with five existing canopy

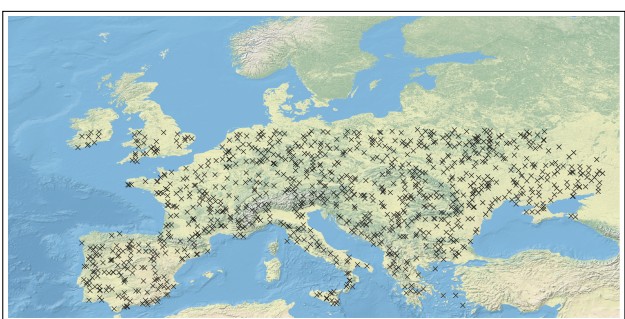

*Figure 2.* Spatial distribution of 1,500 randomly selected validation locations across Europe. Each location covers a $2.56\,\mathrm{km} \times 2.56\,\mathrm{km}$ area in which estimated canopy heights are evaluated against GEDI measurements. Note that due to GEDI's flight path, no labels are available above $51.6°\mathrm{N}$.

height maps (Liu et al., 2023; Tolan et al., 2024; Lang et al., 2023; Turubanova et al., 2023; Pauls et al., 2024), downloaded from Google Earth Engine, and resample all maps to $10\,\mathrm{m}$ in a global projection system (EPSG:4326).

### 4.1. Establishing Baselines

We evaluate our approach by comparing various model configurations. None of these models use explicit time-stamp information (e.g., a separate channel for month or year). Instead, they differ in their treatment of satellite data (e.g., full monthly stacks vs. median composites) and the span of training data (single-year vs. multi-year). In all configurations, Sentinel-1 data – aggregated into a median composite – is concatenated along the channel dimension.

**C1: Single-Year Models Applied to Multiple Years.** In this configuration, a model is trained on data from a single reference year and used to predict canopy height in other years without any retraining. We focus on 2020 due to the availability of published models for this period (Lang et al., 2023; Tolan et al., 2024; Turubanova et al., 2023; Pauls et al., 2024), enabling direct comparisons. We explore the following three variants of this approach:

- 2D-STACK-2020: A 2D U-Net trained on the full 12-month Sentinel-2 stack of 2020, where monthly channels are flattened into a single tensor of shape $[B, H, W, 12 \times \text{Channels}]$.

- 2D-COMPOSITE-2020: A 2D U-Net trained on the median composite of the 12 monthly images, reducing each year's input to a single image of shape $[B, H, W, \text{Channels}]$.

- 3D-STACK-2020: A 3D U-Net (similar to our model) trained solely on the 2020 dataset, using the 12-months stack as a spatio-temporal volume $[B, H, W, 12, \text{Channels}]$.

*Table 2.* Comparison of various model configurations for 2020. The 'STACK' models outperform the 'COMPOSITE' models, and integrating 3D features further boosts performance. Overall, the 3D-STACK-MULTIYEAR model achieves the best results.

|  | MAE [m] | MSE [m$^2$] |
|---|---|---|
| 2D-COMPOSITE-2020 | 5.66 | 85.93 |
| 2D-COMPOSITE-MULTIYEAR | 5.43 | 82.70 |
| 2D-STACK-2020 | 5.51 | 81.31 |
| 2D-STACK-MULTIYEAR | 5.17 | 78.07 |
| 3D-STACK-2020 | 5.48 | 80.32 |
| **3D-STACK-MULTIYEAR** | **5.05** | **77.40** |

After training on the 2020 dataset, we evaluate the 2D-STACK-2020, 2D-COMPOSITE-2020, and 3D-STACK-2020 models on 2019-2022 satellite imagery to assess their generalization without multi-year training.

**C2: Single-Year Models for Each Year.** This configuration trains independent models for each target year. We develop three variants per year—2D-STACK-YEAR, 2D-COMPOSITE-YEAR, and 3D-STACK-YEAR—with each model trained and validated solely on data from its respective year. This approach evaluates how well specialized, single-year models capture inter-annual variability.

**C3: Multi-Year Models.** To enhance generalizability across years, we train models on multi-year data (2019–2022). We use both stack-based (2D-STACK-MULTIYEAR, 3D-STACK-MULTIYEAR) and composite-based (2D-COMPOSITE-MULTIYEAR) approaches. While the 2D-STACK architectures process multi-month data by concatenating all months into a single input dimension, they do not explicitly exploit temporal relationships.

### 4.2. Quantitative Evaluation

We begin by comparing our different model configurations for the reference year 2020 across two different metrics: Mean Absolute Error (MAE) and Mean Squared Error (MSE). As the resulting map is targeted towards forest assessment, we compare it only against labels exceeding $7\,\mathrm{m}$, to exclude labels over acres and grassland. Table 2 excludes 2D-COMPOSITE-YEAR, 2D-STACK-YEAR, and 3D-STACK-YEAR to avoid redundancy, as 2D-COMPOSITE-YEAR is the same as 2D-COMPOSITE-2020 for the 2020 comparison.

Another key comparison is how the metrics vary across different years. Table 3 presents the MAE for each year for all model configurations, as well as the average MAE over all years. The results demonstrate that 3D-STACK-MULTIYEAR significantly outperforms all other configu-

*Table 3.* MAE comparison over multiple years for all model configurations. The 3D-STACK-MULTIYEAR model consistently achieves the lowest MAE. Within each variant (2D-COMPOSITE, 2D-STACK, 3D-STACK), multi-year training substantially outperforms single-year training. Notably, the best single-year model (3D-STACK-2020) only matches the performance of the weakest multi-year model (2D-COMPOSITE-MULTIYEAR).

|  | 2019 | 2020 | 2021 | 2022 | Avg. |
|---|---|---|---|---|---|
| 2D-COMPOSITE-2020 | 6.53 | 5.66 | 5.97 | 5.93 | 6.02 |
| 2D-COMPOSITE-YEAR | 6.08 | 5.66 | 5.83 | 5.64 | 5.80 |
| 2D-COMPOSITE-MULTIYEAR | 5.76 | 5.43 | 5.36 | 5.39 | 5.48 |
| 2D-STACK-2020 | 5.65 | 5.51 | 5.38 | 5.48 | 5.50 |
| 2D-STACK-YEAR | 5.31 | 5.51 | 5.28 | 5.12 | 5.31 |
| 2D-STACK-MULTIYEAR | 5.18 | 5.17 | 4.93 | 4.90 | 5.04 |
| 3D-STACK-2020 | 5.55 | 5.48 | 5.25 | 5.34 | 5.41 |
| 3D-STACK-YEAR | 5.32 | 5.48 | 5.23 | 4.96 | 5.25 |
| **3D-STACK-MULTIYEAR** | **5.09** | **5.05** | **4.84** | **4.83** | **4.95** |

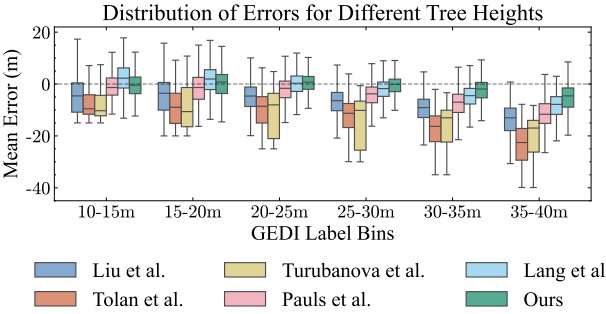

*Figure 3.* Boxplots for each model showing the 2020 mean error in every 5 m bin between 10 m and 40 m. Although Liu et al. (2023), Pauls et al. (2024) and Lang et al. (2023) perform well on smaller trees, our models performs especially well for taller trees.

rations, both in single-year and multi-year analyses. In contrast, 2D-COMPOSITE consistently shows the poorest performance, followed by 2D-STACK and 3D-STACK.

### 4.3. Method Comparison

As outlined in Section 3.3, all models in Table 2 and Table 3 were trained on a smaller dataset. After identifying the best performing configuration, we trained a new model, 3D-STACK-MULTIYEAR-L, on the full dataset, which we will refer to as *Ours* from this point forward. Unless stated otherwise, all values in this section are based on labels greater than 7 m to focus on forest-related data.

**Quantitative Evaluation.** Table 4 reports MAE, MSE, and coefficient of determination $R^2$ for five maps: a global map at 1 m resolution (Tolan et al., 2024), a European map at 3 m resolution (Liu et al., 2023)[7], global maps at 10 m resolution (Lang et al., 2023; Pauls et al., 2024), and a

---

[7]Liu et al. (2023)'s 2019 map is included for completeness.

*Table 4.* Comparison of performance metrics for different models in 2020. Despite the coarser 10 m resolution of Sentinel-1/2 (S1/2) compared to Planet (3 m) and Maxar (60 cm), our model yields highly accurate maps and achieves best overall performance.

|  | Source | MAE [m] | MSE [m²] | $R^2$ |
|---|---|---|---|---|
| Tolan et al. (2024) | Maxar | 11.25 | 212.14 | 0.409 |
| Liu et al. (2023) | Planet | 8.17 | 138.25 | 0.481 |
| Lang et al. (2023) | S2 | 5.74 | 84.68 | 0.488 |
| Pauls et al. (2024) | S1/2 | 5.46 | 83.14 | 0.536 |
| Turubanova et al. (2023) | Landsat | 12.39 | 252.57 | 0.318 |
| **Ours** | **S1/2** | **4.76** | **74.28** | **0.591** |

*Table 5.* Direct MAE comparison across different years between our model and Turubanova et al. (2023). On average, our model improves predictions by 61%, with notable year-to-year variability (reflected in a standard deviation of $\sigma = 0.27$ m).

|  | 2019 | 2020 | 2021 | 2022 | **Avg. (all years)** |
|---|---|---|---|---|---|
| Turubanova et al. | 12.38 | 12.39 | 11.37 | – | 12.05 (–) |
| **Ours** | **4.77** | **4.76** | **4.53** | **4.48** | **4.69 (4.64)** |

European map at 30 m resolution (Turubanova et al., 2023).

Despite the coarser resolution of Sentinel-1/2 (10 m) compared to Planet (3 m) or Maxar (60 cm), models utilizing Sentinel data achieve higher accuracy, likely due to the availability of near-infrared and shortwave-infrared bands. As shown in Table 4, our model, 3D-STACK-MULTIYEAR-L, achieves the best performance across all metrics, with an MAE of 4.76 m—representing a 13% improvement over the next-best model (Pauls et al., 2024). Similar improvements are observed for MSE and $R^2$.

Turubanova et al. (2023) offers the only other high-resolution, multi-year tree canopy height map trained at European scale, with predictions spanning 2001–2021. Table 5 compares the MAE of their map for 2019–2021 with the performance of our model, 3D-STACK-MULTIYEAR-L, evaluated from 2019–2022. Our model consistently outperforms Turubanova et al. (2023) across all shared years, achieving MAE values less than half theirs (61% improvement). Although both models exhibit notable inter-annual variations, likely due to factors such as lighting conditions, forest dynamics, and differences in label distributions, the results demonstrate the superior accuracy and robustness of our model for European tree canopy height estimation.

Tree canopy height maps often suffer from reduced accuracy as tree height increases; a serious problem as tree canopy height maps are often used for above-ground biomass prediction, where biomass typically scales non-linearly with height following an allometric power-law relationship. Figure 5 shows the mean error within 5 m bins between 10 m and 40 m, with a negative mean error denoting that the prediction is on average lower than the label. For smaller trees

| Google Maps | Ours | Lang et al. | Liu et al. | Tolan et al. | Pauls et al. | Turubanova et al. |
|---|---|---|---|---|---|---|

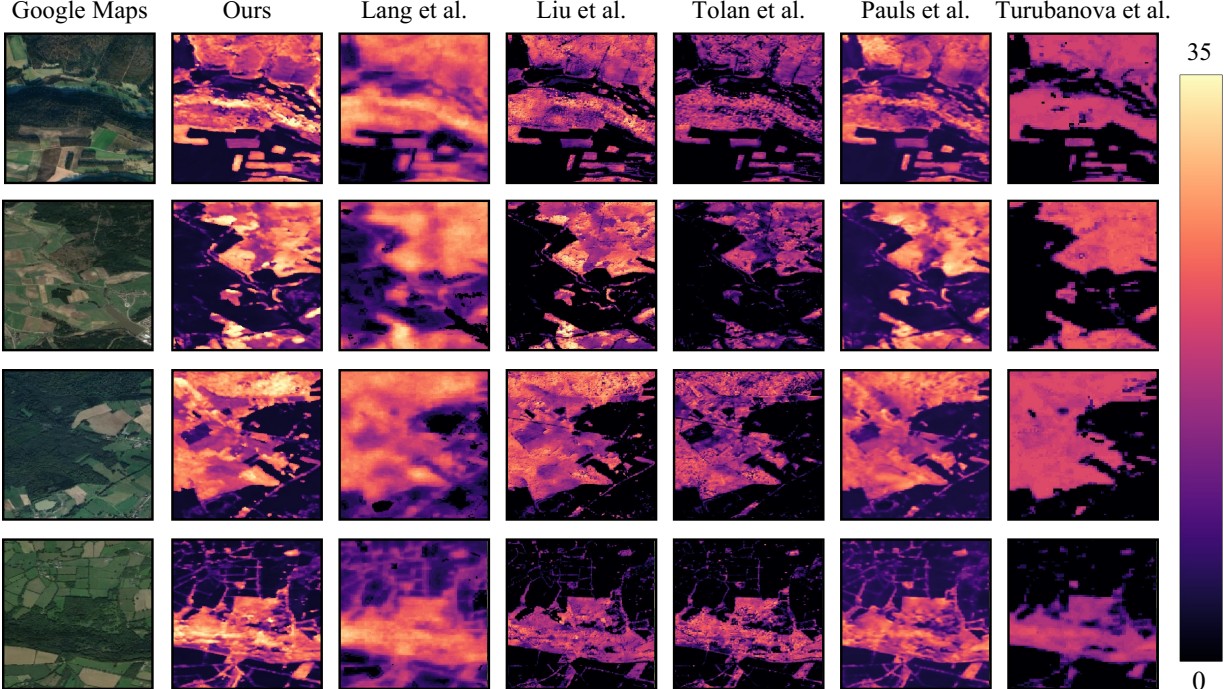

*Figure 4.* Qualitative comparison of canopy height maps for the reference year 2020: Liu et al., Tolan et al., Lang et al., Pauls et al., Turubanova et al. (2020) and our model, 3D-STACK-MULTIYEAR-L.

ranging from $10\,\text{m}$ to $20\,\text{m}$, Liu et al. (2023); Pauls et al. (2024); Lang et al. (2023) and our model perform reasonably well, whereas Tolan et al. (2024) and Turubanova et al. (2023) show mean errors around $-10\,\text{m}$, which worsen further for taller trees. Notably, from $20\,\text{m}$ onward, our model outperforms all others, achieving a mean error of approximately $-5\,\text{m}$ between $35\,\text{m}$ and $40\,\text{m}$, significantly reducing the error compared to the next best model by Lang et al., particularly for these ecologically critical tall trees. This substantial improvement highlights our model's ability to provide more reliable tree canopy height estimates for tall trees, which are key contributors to carbon storage.

Figure 5 shows a scatterplot for each model along with its $R^2$ value, where $R^2$ (coefficient of determination) quantifies the proportion of variance in the tree canopy height labels explained by the model predictions (a 1 corresponds to a perfect fit, a 0 means that no variance is explained). Here, $R^2$ is calculated using all labels and $R_7^2$ takes only labels exceeding $7\,\text{m}$. The plots reveal that Tolan et al. (2024) and Turubanova et al. (2023) struggle with taller trees, as neither predict heights beyond $25\,\text{m}$. Lang et al. (2023), Liu et al. (2023), and Pauls et al. (2024) tend to saturate at $35\,\text{m}$. In contrast, our model exhibits the narrowest point cloud, with the highest density areas aligning closely with the perfect fit line, and achieves strong performance up to $40\,\text{m}$–$45\,\text{m}$. Consequently, we improve $R^2$ from 0.793 to 0.819, and for labels over $7\,\text{m}$ ($R_7^2$), from 0.536 to 0.591.

**Qualitative Evaluation.** While quantitative evaluation is essential, visual quality is equally important for tree canopy height prediction. Figure 4 visually compares Liu et al. (2023), Tolan et al. (2024), Lang et al. (2023), Pauls et al. (2024), Turubanova et al. (2023) and our model. The first column shows a high-resolution satellite image from Google Maps (not necessarily from 2020) whereas all other columns show the height between $0\,\text{m}$ (black) and $35\,\text{m}$ (light yellow). Despite its relatively modest metrics, the high resolution of Tolan et al. (2024)'s map makes it effective for canopy detection. Liu et al. (2023) identifies trees with higher resolution and does not suffer as much from saturation effects as Tolan et al. (2024) and Turubanova et al. (2023). While our model lacks the high resolution of Liu et al. (2023) and Tolan et al. (2024), it surpasses Pauls et al. (2024), Lang et al. (2023), and Turubanova et al. (2023). Unlike other maps that assign similar heights to all forest patches, our map is the only one differentiating various forest heights in smaller patches (bottom of first row).

One strong improvement of our model is the detection of forest patches with very high trees. Figure 1 shows a small, but representative example of a forest with patches of high trees, where we have very high-quality ALS data. Although Liu et al. (2023) and Pauls et al. (2024) detect that these trees are at a different height than their surrounding, they fail to correctly estimate the height. Our model is able to detect those patches and accurately predict its height.

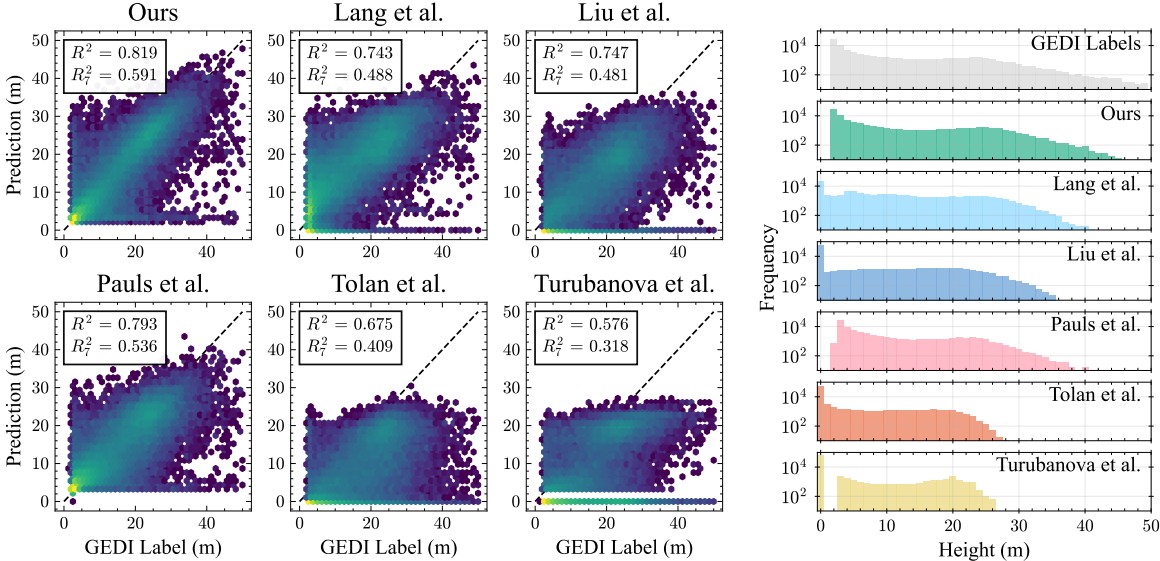

*Figure 5.* Left: Scatterplots between 2020 GEDI labels and prediction for Lang et al.; Liu et al.; Pauls et al.; Tolan et al.; Turubanova et al. and our model including $R^2$ for all labels and $R_7^2$ for labels exceeding $7\,\mathrm{m}$. Right: Histograms of GEDI labels and all maps. Turubanova et al. and Tolan et al. saturate at $28\,\mathrm{m}$, our model is the only one matching above $40\,\mathrm{m}$.

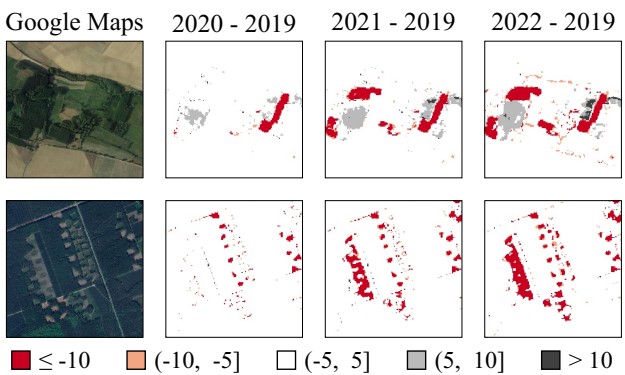

*Figure 6.* Temporal maps illustrate the expansion of deforestation from 2019 to 2022. This is observed by comparing differences between each year, visible in both solitary forest patches surrounded by open land and within densely forested areas.

As our model is applied to 2019-2022, we can observe temporal dynamics such as deforestation, identifying where trees have been cut down. Figure 6 shows observable deforestation in two areas. However, mapping minor tree growth is challenging due to the short time frame and slow natural growth rate of trees. In contrast, in forest plantations where fast-growing tree species are actively managed, significant growth can be observed more easily because the acceleration in growth outweighs the uncertainty in our predictions.

## 5. Conclusion

We presented a novel approach for generating high-resolution, large-scale temporal tree canopy height maps using a 3D U-Net model. By utilizing a full 12-month Sentinel-2 imagery time series, our approach avoids the information loss of median composites and captures essential seasonal variations and geolocation shifts. Trained on GEDI LiDAR data, our model produces a highly accurate $10\,\mathrm{m}$ resolution tree canopy height map of Europe for the years 2019 to 2022. Our model outperforms existing state-of-the-art models, achieving significantly lower errors. Notably, it drastically improves accuracy for tall trees, essential for precise biomass estimation and carbon stock assessments. Our temporal maps capture forest dynamics, such as deforestation and growth patterns, offering valuable insights for forest monitoring and ecological analyses. We believe that the publicly released pipeline and tree canopy height maps will support further research and inform decision-making in forest management and conservation.

## Acknowledgements

This work was supported via the AI4Forest project, which is funded by the German Federal Ministry of Education and Research (BMBF; grant number 01IS23025A) and the French National Research Agency (ANR). It was also partially supported by the *Deutsche Forschungsgemeinschaft* (DFG, German Research Foundation) under Germany's Excellence Strategy—The Berlin Mathematics Research Center MATH+ (EXC-2046/1, project ID: 390685689). We also acknowledge the computational resources provided by the PALMA II cluster at the University of Münster (subsidized by the DFG; INST 211/667-1) as well as by the Zuse Institute Berlin. We also appreciate the hardware donation of an A100 Tensor Core GPU from Nvidia and thank Google for their compute resources provided (Google Earth Engine).

## Impact Statement

This study significantly advances forest monitoring by leveraging machine learning to incorporate temporal dynamics, resulting in the creation of tree canopy height maps over a four-year span (2019–2022) on a continental scale. Forests play a vital role in mitigating climate change by absorbing nearly half of human-generated carbon dioxide emissions (Friedlingstein et al., 2022). However, they are increasingly at risk due to climate change and human activities such as deforestation and land degradation (Anderegg et al., 2022). In support of global initiatives like the United Nations' Sustainable Development Goals (SDGs)[8], the Bonn Challenge[9], and the Glasgow Declaration[10], effective forest management and conservation are crucial for climate adaptation and mitigation. Our approach significantly benefits the estimation of above-ground biomass, a critical component among the 55 essential climate variables (ECVs)[11]. A key improvement in this research is the enhanced detection of large trees, which is critical due to the power-law relationship between tree height and biomass. This advancement allows for more accurate biomass estimation, essential for assessing carbon stocks. Additionally, precise monitoring is crucial for validating carbon credit investments in the voluntary carbon market, ensuring the effectiveness and sustainability of forest growth projects while reducing potential leakage effects.

---

[8]https://sdgs.un.org/2030agenda
[9]https://www.bonnchallenge.org
[10]https://www.oneplanetnetwork.
org/programmes/sustainable-tourism/
glasgow-declaration
[11]https://gcos.wmo.int/site/
global-climate-observing-system-gcos/
essential-climate-variables/
above-ground-biomass

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

## A. Data Handling Details

Handling big amounts of data is a challenge. To aid data handling and still having optimal projection accuracy, we downloaded, stored and preprocessed all data in the original Sentinel-2 tiling system. The tiling system is based on the UTM projections system but divides every UTM zone in $100\,\text{km} \times 100\,\text{km}$ tiles. Each UTM zone has its own projection[12].

Sentinel-1 images are created with Google Earth Engine (GEE) [13] directly in their corresponding projection and compressed with LZW (the standard compression in GEE). Sentinel-2 images are distributed via the Copernicus AWS via a .SAFE-archive format. We extract all needed bands in JPEG2000-format and build virtual raster files for combined access. JPEG2000 has the big advantage of being an extremely storage efficient, yet applies the compression on image-level, so windowed access is not possible.

Each sample of the final dataset is stored in a zip-compressed archive to further save storage.

When doing inference, directly loading the entire tile and doing inference is not feasible, as data loading takes a long time due to the need to be decompressed. As this would lead to long GPU idle times, we use a different approach. We separate the workload onto CPU- and GPU-nodes, where CPU-nodes load the data from the slow storage, decompress it multi-threaded and save it in a binary format on fast-access storage. The GPU-node then loads the fast binary and does inference. For each GPU-worker we start multiple CPU-workers to remove GPU idle time.

## B. Ablation Studies

In the main paper we took several decisions without justification. Here we provide experimental ablation studies to underpin these conclusions.

### B.1. Additional Sentinel-2 Bands

Sentinel-2 captures bands in resolutions of $10\,\text{m}$, $20\,\text{m}$ and $60\,\text{m}$ and in this study we included all (except for B10, which is only for clouds). However, both $60\,\text{m}$ bands have wavelength, where direct knowledge transfer to this application is unknown: B01 measures coastal aerosols and B09 captures the density of water vapors. Our experiments show that both bands do not significantly increase the performance, neither do they decrease the performance. Further analysis is needed, but both bands are candidates to be removed from the set of input channels. Table 6 reports the results on the validation part of our dataset (L1 > 15 m refering to the L1 loss for all labels that exceed 15 m)

*Table 6.* Comparison of error metrics with and without spectral bands B01 and B09.

| Configuration | L1 (m) | L1 > 15 m (m) | L1 > 20 m (m) | L1 > 25 m (m) | L1 > 30 m (m) | L2 (m) |
|---|---|---|---|---|---|---|
| Without B01 and B09 | 1.991 ± 0.002 | 4.837 ± 0.008 | 5.476 ± 0.010 | 7.384 ± 0.008 | 11.406 ± 0.004 | 22.281 ± 0.037 |
| Including B01 and B09 | 1.992 ± 0.003 | 4.830 ± 0.014 | 5.460 ± 0.015 | 7.364 ± 0.029 | 11.384 ± 0.031 | 22.277 ± 0.053 |

### B.2. Time-Series: Seasonal pPatterns

Although our main paper shows that a time-series clearly outperforms the baseline setting of applying a median aggregation to the input data, we want to dive deeper in how the model leverages e.g. differences across the monthly images. We present 3 experiments: activation patterns for different months, performance for different forest types (with different phenological patterns) and an ablation between using only summer/winter months or both.

#### B.2.1. ACTIVATION PATTERNS FOR DIFFERENT MONTHS

We use Guided Attention (Springenberg et al., 2014) to visualize the activation patterns across months for different patches (cf. Figure 7. We observe varying activation strengths across months and patches, suggesting the model processes temporal information differently by location. However, further research would be needed to confirm this hypothesis.

#### B.2.2. PERFORMANCE ON DIFFERENT FOREST TYPES

---

[12]For more details, we refer to: https://sentiwiki.copernicus.eu/web/s2-products
[13]https://earthengine.google.com

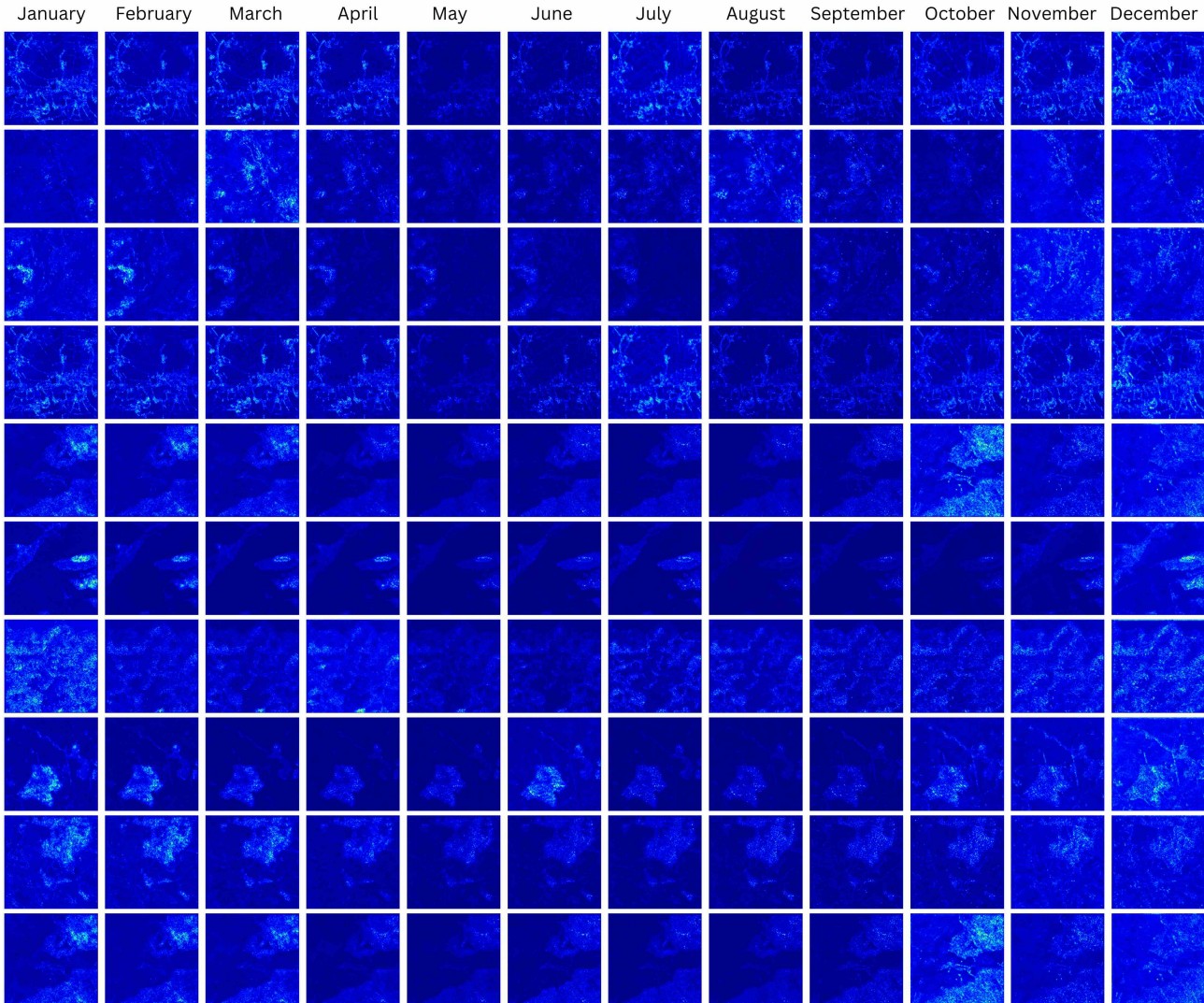

*Figure 7.* The activation patterns differ across months signaling that some images are more important, i.e. contain more information, than others. Further, there is no pattern between multiple patches.

We evaluated our model separately on broadleaf and coniferous forests using the Copernicus Land Monitoring Service Forest Type Map (2018) (Copernicus Land Monitoring Service, 2021). These forest types show different seasonal patterns - broadleaf forests have distinct leaf-on/off periods while coniferous forests maintain constant canopy. Table 7 reports our findings.

*Table 7.* Comparison across tree types for different models.

| MODEL | BROADLEAF MAE (M) | CONIFEROUS MAE (M) |
|---|---|---|
| LANG ET AL. | 5.44 | 5.11 |
| LIU ET AL. | 7.01 | 6.91 |
| PAULS ET AL. | 5.30 | 4.85 |
| TOLAN ET AL. | 10.43 | 11.73 |
| TURUBANOVA ET AL. | 8.60 | 8.43 |
| **OURS** | **4.57** | **4.11** |

### B.2.3. SEASONAL INFORMATION VARIATION

To investigate the benefits of seasonal information independently from data volume (Note: the number of training labels remains identical for all variants), we conducted an ablation study comparing three models trained on different 4-month subsets: Winter (Nov-Feb), Summer (Jun-Sep) and Mixed (Jan-Feb, Aug-Sep).

The results in Table 8 show that using only summer/leaf-on months is superior to using only winter/leaf-off months, however a mix of winter and summer months yields better validation performance than both individually.

*Table 8.* Seasonal variation in performance using Huber Loss (in meters) for different model variants.

| MODEL VARIANT | HUBER LOSS (M) |
|---|---|
| WINTER (NOV–FEB) | $1.169 \pm 0.003$ |
| SUMMER (JUN–SEP) | $1.130 \pm 0.002$ |
| MIXED (JAN–FEB, AUG–SEP) | $1.122 \pm 0.002$ |

## C. Analysis

Although our primary goal was to develop an openly available model that others can use for their own analyses, here we demonstrate the model's temporal capabilities. We analyzed potential deforestation events by tracking pixels for which the corresponding height decreased from above 8m to below 5m between years. The affected area increased from $9747.9 \text{ km}^2$ between 2019 and 2020 and $7729.1 \text{ km}^2$ between 2020 and 2021 to $15942.5 \text{ km}^2$ between 2021 and 2022.

# D. Additional Figures

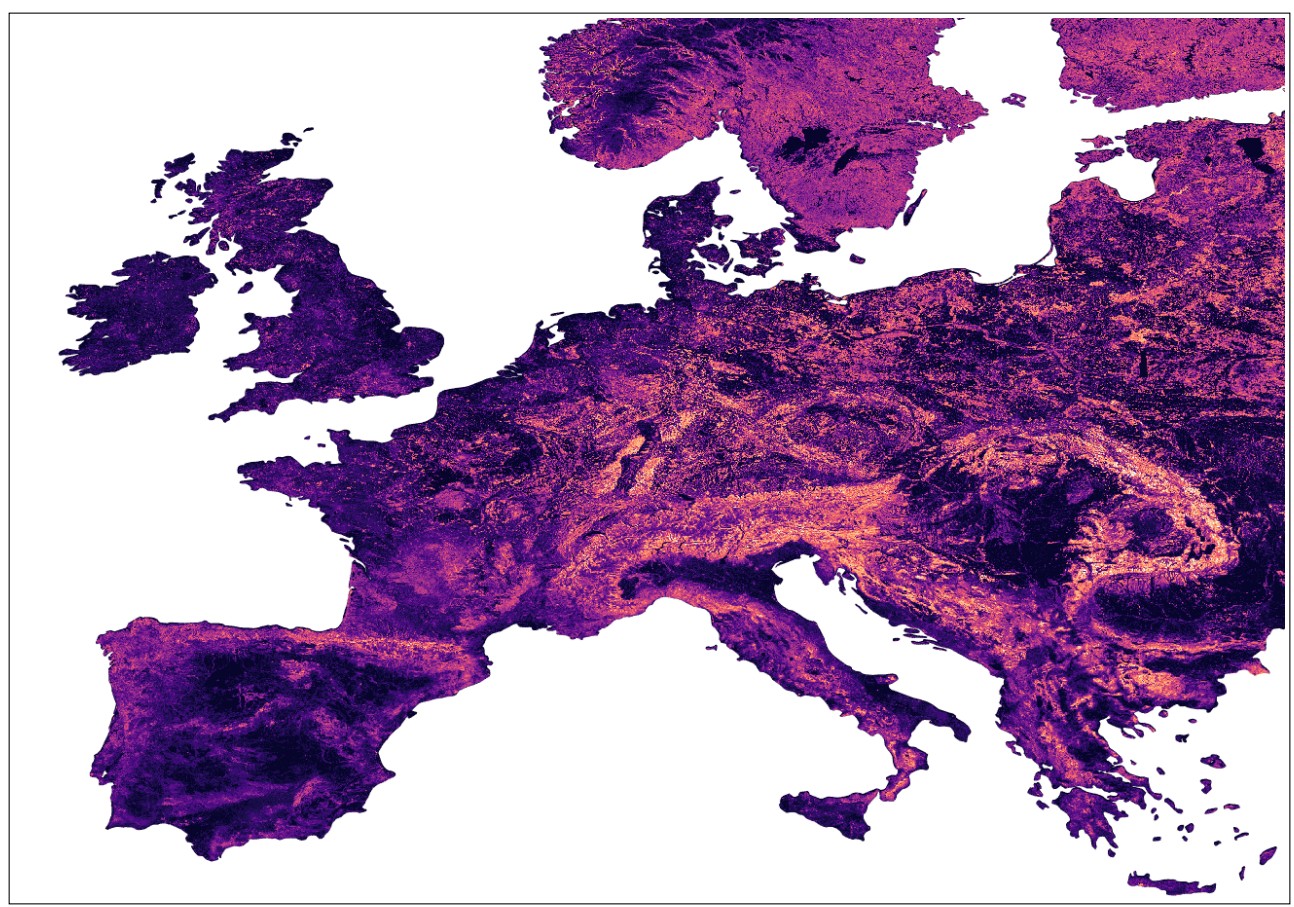

*Figure 8.* Tree canopy height map for central Europe for 2020.

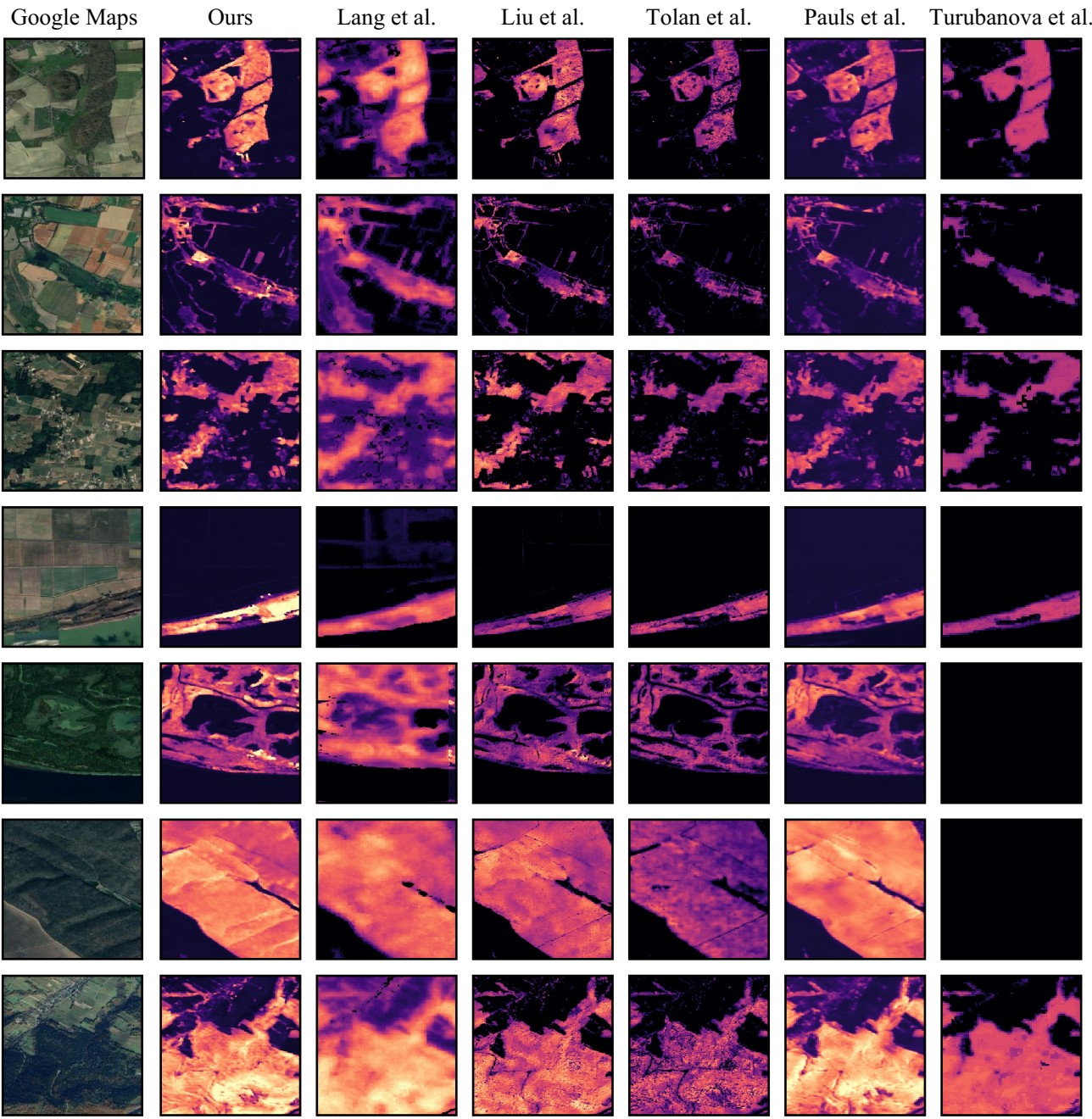

*Figure 9.* Comparison of different tree canopy height products, including Lang et al. (2023); Liu et al. (2023); Tolan et al. (2024); Pauls et al. (2024); Turubanova et al. (2023). The first column shows a high-resolution image from Google Maps.

