# OpenReview forum: "Capturing Temporal Dynamics in Large-Scale Canopy Tree Height Estimation"
_ICML.cc/2025/Conference — ICML 2025 poster_

### Official Review · Reviewer_edjr · 2025-03-11

**Overall Recommendation:** 5

**Summary:**

This paper is the first to produce a 10m resolution time-series forest height map of Europe from 2019 to 2022. The 2020 results were compared with multiple tree height studies, revealing that its accuracy is also the most reliable. The data used include GEDI, Sentinel-1, and Sentinel-2, with the model being 3D-UNet, the loss function Huber Loss, and the Adam optimizer. The paper also presents findings on temporal height changes, such as tree felling.

**Claims And Evidence:**

Claims (such as first 10m-time-serises map  and better accuracy) in this work are well supported.

**Essential References Not Discussed:**

No

**Experimental Designs Or Analyses:**

I have checked the soundness of the experimental design and analysis. I found that the experiment itself is well-structured and complete. However, the analysis section needs further improvement. For example, there is a lack of temporal accuracy analysis, phenomenon discovery, and discussion. Currently, it seems that only Figure 6 and Tab 5 present these aspects, whereas the key highlight of the paper is the temporal aspect.

**Methods And Evaluation Criteria:**

Yes.  Make sense for the application.

**Other Comments Or Suggestions:**

N/A

**Other Strengths And Weaknesses:**

As a machine learning application in geosciences, this work is well-structured and comprehensive, with insightful results, which is a strong point.

However, a key weakness is that the study primarily applies existing models and algorithms without introducing innovations in machine learning methods or techniques. For instance, why did previous static inversion methods fail in this task, while this study, using the same or similar existing algorithms, succeeds? Is it due to differences in data input, or does it stem from how the time-series data is processed and integrated? A deeper analysis of these aspects would strengthen the contribution.

I agree that innovation can be based on existing methods, but it is essential to explain why the current approach, without major modifications (such as in the loss function or data mining paradigm), is able to achieve what previous methods (e.g., single-year inversion) could not. This is the truly insightful aspect, rather than merely an engineering application.

**Questions For Authors:**

[1] If the improved results are merely due to the fact that previous methods aggregated the data while this study processes them separately, the authors should explicitly state this.

[2] The model structure mentions, "12 monthly Sentinel-2 images concatenated with an aggregated Sentinel-1 composite." So, what is the total number of input channels? Since a year's worth of data is extremely large and contains various uncertainties (such as clouds and other artifacts), how are these issues handled?

[3] Why is the 'Sentinel-1 data aggregated into a single median-based image over the year' ? Is it because the data lacks clear periodicity?

[4]  The model's training configuration, duration, and computational resources should be provided in detail.

[5] If the reported MAE in this paper is 4.64m, what is the accuracy of GEDI? What is GEDI's temporal accuracy? Is the accuracy upper bound for time-series forest height inversion determined by GEDI's accuracy?

[6] Can you provide some failed cases and explain why the current method cannot handle them? Additionally, for low vegetation (below 5 meters), does this study face challenges in height estimation and time-series measurement?

**Relation To Broader Scientific Literature:**

Forest height is critically important, and time-series forest height is a key indicator for assessing Earth's health and estimating carbon sink biomass. Currently, most studies focus on static forest height estimation, while dynamic inversion remains scarce. The high-resolution time-series inversion proposed in this paper fills this gap.

**Theoretical Claims:**

No theoretical proof in this work.

---

> ### Author Rebuttal · Authors · 2025-04-01
>
> Thank you for your thorough review and for acknowledging our work. Let us address your remarks and questions in detail below.
>
> > However, the analysis section needs further improvement. [...]
>
> We agree that this is an important aspect. Our primary goal was to develop an openly available model that others can use for their own analyses. To demonstrate the model's temporal capabilities, we analyzed potential deforestation events by tracking pixels for which the corresponding height decreased from above 8m to below 5m between years. The affected area increased from 7,729.1 km² in 2020-2021 to 15,942.5 km² in 2021-2022, also see our answer to reviewer 1141. We have updated our manuscript accordingly
>
> > However, a key weakness is that the study primarily applies existing models and algorithms [...]
> >
> > [1] If the improved results are merely due to [...].
>
> We would like to emphasize that our work follows the application-driven track. While pinpointing the exact source of improvement is challenging due to significant architectural differences, the results from Table 2 demonstrate three key improvements:
>
> 1. Utilizing 12 monthly images rather than a single composite image (`-Composite-` vs `-Stack-`) reduces MAE by 3.6%
> 2. Processing temporal data with 3D convolutions instead of 2D convolutions (`2D-Stack-` vs `3D-Stack-`) further reduces MAE by 1.4%
> 3. Training on multiple years of data (2019-2022) versus a single year (2020) (`-Year` vs `-MultiYear`) reduces MAE by 6%
>
>
> > [2] The model structure mentions, "12 monthly Sentinel-2 images concatenated with an aggregated Sentinel-1 composite." [...]
>
> The model receives 12 monthly images, each containing 16 channels - the Sentinel-2 bands for that month concatenated with the yearly median of Sentinel-1 data. So technically, the model overall receives 12*16=192 channels as input, although we do not concatenate these, but rather process them in the 3D-U-Net as (12, 16, 256, 256) image input. Regarding your second question, we do not explicitly filter clouds from the input images, but the model demonstrates robust performance even with partially cloudy inputs, maintaining high prediction accuracy. We utilize the L2A product, which includes atmospheric correction and other preprocessing steps.
>
> > [3] Why is the 'Sentinel-1 data aggregated into a single median-based image over the year' ? Is it because the data lacks clear periodicity?
>
> While Sentinel-1 has similar temporal resolution to Sentinel-2, we opted to use composite radar data since it reduces noise and speckle artifacts through temporal aggregation, while keeping computational costs manageable. In contrast, we found using the full temporal sequence beneficial for optical Sentinel-2 data.
>
> > [4] The model's training configuration, duration, and computational resources should be provided in detail.
>
> In Section 3.3 of the paper, we provide detailed information about the experimental setup, including the considered optimizer, learning rate, batch size, and more. Regarding duration and computational resources, we trained the model for a duration of roughly four days on two A100 GPUs. We will add detailed information about the computational resourced used to the revision. Thank you very much for this remark.
>
>
> > [5] If the reported MAE in this paper is 4.64m, what is the accuracy of GEDI? What is GEDI's temporal accuracy? Is the accuracy upper bound for time-series forest height inversion determined by GEDI's accuracy?
>
> GEDI has a measurement resolution of 15cm but varying accuracy by surface type - overall MAE of 0.98m, with higher error in tree-covered areas (1.67m) versus grassland (0.79m) and cropland (0.57m) [PEL]. GEDI cannot reliably measure heights 0-3m and has reduced accuracy below 5m. Regarding temporal accuracy - GEDI rarely measures the same location twice, so it lacks a temporal dimension. For accuracy bounds - as GEDI provides our training labels, this likely sets an upper limit on our model's achievable accuracy. Is this what you meant?
>
> [PEL] Pronk, M., Eleveld, M., & Ledoux, H. (2024). Assessing vertical accuracy and spatial coverage of ICESat-2 and GEDI spaceborne lidar for creating global terrain models. Remote Sensing, 16(13), 2259.
>
> > [6] Can you provide some failed cases and explain why the current method cannot handle them? Additionally, for low vegetation (below 5 meters), does this study face challenges in height estimation and time-series measurement?
>
> We provide two failure cases: 1) GEDI measurements on slopes, where terrain changes are mistaken for vegetation height differences, and 2) checkerboard artifacts in harbor areas. We will expand this analysis in the appendix. Additionally, as noted earlier, both GEDI and our model struggle with accurate measurements of low vegetation.
>
>
> We hope to have addressed all your concerns, please let us know if further clarification is needed. Thank you!

---

> > ### Comment · Reviewer_edjr · 2025-04-03
> >
> > The dataset and the research itself are highly meaningful, and I believe the work could have a significant impact across multiple disciplines and research areas.
> >
> > Your detailed response has addressed most of my concerns, and I believe the paper can be raised to  Accept.
> >
> >
> > I still have a few minor questions:
> >
> > 1. Inspired by Reviewer sUWg, I’m curious—within your framework, what are the key differences between building and vegetation height inversion? Since vegetation height can be supervised using GEDI tree height products (no building-height product), is the only difference between building and vegetation height inversion just the source of ground truth?
> >
> > 2. You mentioned accuracy seems to refer to GEDI's observation precision. However, your tree height labels are from GEDI products, which are already processed, right? Is their accuracy consistent with the raw GEDI observations? Or did I misunderstand something?
> >
> > 3. It seems that “Training on multiple years of data (2019–2022) versus a single year (2020) (-Year vs -MultiYear) reduces MAE by 6%” contributed the most to performance improvement.  So I’d like to confirm if my understanding is correct:
> >
> >     a) The version trained with fewer labels (only one year) actually achieved better accuracy. In other words, for this application, mixing multi-year labels may introduce inconsistency or noise, **so fewer but more consistent (high signal-to-noise ratio) labels are more beneficial.**
> >
> >
> >     b) If that’s the case, **would it be more effective to use fewer but higher-quality labels**—for example, sampling only 1/4 of high-quality labels each year (e.g., non-cloud, high-quality waveform/tree height)—**to construct a cleaner, multi-year training set?**
> >
> > Given time constraints, **I don't expect new experiments**, but I’d like to confirm whether I understood this correctly, and whether this is also how the authors interpret the results. If not, please feel free to correct me—or perhaps consider discussing this point further in a revised revision.

---

> > > ### Author Response · Authors · 2025-04-08
> > >
> > > Thank you for your feedback and further questions!
> > >
> > > > Inspired by Reviewer sUWg, I'm curious—within your framework, what are the key differences between building and vegetation height inversion? Since vegetation height can be supervised using GEDI tree height products (no building-height product), is the only difference between building and vegetation height inversion just the source of ground truth?
> > >
> > > Yes, the only difference lies in the source of ground-truth data. While GEDI is designed to measure vegetation height, it still captures height measurements in urban areas, which explains the height variations we observe in cities. However, since GEDI is not optimized for building measurements, these urban height estimates are less reliable than vegetation measurements; the focus of our work lies on tree canopy height estimation.
> > >
> > > > You mentioned accuracy seems to refer to GEDI's observation precision. However, your tree height labels are from GEDI products, which are already processed, right? Is their accuracy consistent with the raw GEDI observations? Or did I misunderstand something?
> > >
> > > We are not entirely sure if we understand correctly. Yes, the GEDI labels we use for training are already "pre-processed" by the GEDI provider in the sense that GEDI actually returns a waveform, hence a 1-dimensional array of photon information. The label we use corresponds to the L2A product, which just returns the "rh98" value, which is the 98th percentile of the waveform, i.e. the height value such that 98% of the photons returned are below this value (cf. lines 186). In that sense, this is entirely consistent with the raw GEDI observation, it is however just a single statistic of the waveform. Does this answer your question? If not, please let us know.
> > >
> > > > It seems that "Training on multiple years of data (2019–2022) versus a single year (2020) (-Year vs -MultiYear) reduces MAE by 6%" contributed the most to performance improvement. So I'd like to confirm if my understanding is correct:
> > > a) The version trained with fewer labels (only one year) actually achieved better accuracy. In other words, for this application, mixing multi-year labels may introduce inconsistency or noise, so fewer but more consistent (high signal-to-noise ratio) labels are more beneficial.
> > > b) If that's the case, would it be more effective to use fewer but higher-quality labels—for example, sampling only 1/4 of high-quality labels each year (e.g., non-cloud, high-quality waveform/tree height)—to construct a cleaner, multi-year training set?
> > > Given time constraints, I don't expect new experiments, but I'd like to confirm whether I understood this correctly, and whether this is also how the authors interpret the results. If not, please feel free to correct me—or perhaps consider discussing this point further in a revised revision.
> > >
> > > We believe there might be a misunderstanding.
> > >
> > > a) Could you elaborate what you mean by "accuracy" here? In Table 2 we report MAE, MSE, and RMSE, hence three metrics where lower is better. Training on just a single year consistently achieves higher errors compared to training on multiple years. Further note that in this table we account for the number of samples throughout training, i.e., training on a single year just uses information from a single year, but not necessarily fewer labels. We will make sure to make this more clear in the paper, if necessary.
> > >
> > > b) If we understand you correctly, point a) is not the case, however we do agree that using higher-quality labels is beneficial. In fact, we are actively investigating this trade-off between label quantity and quality in our ongoing research.
> > >
> > > We hope to have clarified your questions and remain at your disposal for any further questions.

---

### Official Review · Reviewer_qZ94 · 2025-03-12

**Overall Recommendation:** 2

**Summary:**

The paper presents an approach for creating large-scale temporal tree canopy height maps using satellite imagery. The main contributions include a deep learning model (3D U-Net architecture) that can track forest height changes across Europe from 2019-2022 at 10m spatial resolution; a canopy height map of Europe for 2020; and the finding that using full 12-month time series of Sentinel-2 imagery improves performance by capturing seasonal patterns and leveraging geo-location shifts, compared to aggregated composites).

**Claims And Evidence:**

* Capturing seasonal patterns: The paper claims that using full 12-month time series of Sentinel-2 imagery improves performance by "capturing seasonal patterns" (compared to using aggregated composites).
    * The paper lacks direct analysis of how the model actually uses seasonal information, with examples showing different predictions in leaf-on vs leaf-off seasons. While the paper shows that using 12 months performs better than composites, they don't isolate whether this improvement is due to seasonal patterns or simply having more data points.
    * To properly support this claim, the paper should include analysis of the model's attention to seasonal changes, performance comparisons across seasons, and demonstrations of different behavior for deciduous vs evergreen forests. Explicit testing is needed to show that the improvement comes from seasonal information rather than just more data points. Visualizations or examples showing how the model uses seasonal information, along with ablation studies isolating the impact of seasonal patterns, would significantly strengthen the evidence for this claim.


* leveraging geo-location shifts: the paper claims that by processing a stack of 12 monthly Sentinel-2 images concatenated with an aggregated Sentinel-1 composite, the method leverages geolocation offsets in Sentinel-2 imagery.
    * The paper lacks visualization or quantitative analysis demonstrating how the model uses geolocation shifts, including examples showing improved edge detection or fine spatial details that could be attributed to leveraging these shifts. On the contrary the spatial resolution is lesser than Liu et al. The improved performance could be due to other factors like increased data volume or temporal information, rather than specifically leveraging geolocation shifts.
    * To properly support this claim, the paper should include an ablation study isolating the impact of geolocation shifts from other factors, with metrics or measurements showing the degree of improvement that can be attributed to leveraging geolocation shifts. Providing examples showing enhanced edge detection or spatial detail due to this technique will strengthen the claims in this paper.

**Essential References Not Discussed:**

No

**Experimental Designs Or Analyses:**

Yes

**Methods And Evaluation Criteria:**

The methods and evaluation criteria employed in the paper are generally appropriate for the problem of large-scale canopy height estimation, with some notable strengths and limitations:

Strengths:

* Use of GEDI LiDAR data as ground truth aligns with standard practices in the field and comprehensive comparison with existing methods using multiple metrics (MAE, MSE, RMSE, R²)
* Inclusion of both quantitative metrics and qualitative visual comparisons with use of high-quality ALS data for additional validation of tall tree detection

Limitations:

* Temporal validation relies heavily on detecting deforestation, with limited validation of growth detection
* While multiple metrics are used, they don't specifically address the claimed benefits of seasonal patterns and geolocation shifts

**Other Comments Or Suggestions:**

Not Applicable

**Other Strengths And Weaknesses:**

The paper's most significant contribution is making high-resolution temporal forest monitoring more accessible through its integration with Google Earth Engine. While individual technical components might not be groundbreaking, their combination and practical implementation represents applied machine learning for environmental monitoring.

Weaknesses:

* Technical Limitations:
    * Lack of detailed analysis supporting claims about seasonal patterns
    * Insufficient evidence for geolocation shift benefits
    * Limited validation of temporal dynamics, especially for forest growth
* Practical Implementation Details:
    * Limited discussion of computational requirements
    * Minimal discussion of model robustness to different environmental conditions

**Questions For Authors:**

1. Seasonal Pattern Analysis: How does the model specifically utilize seasonal information? Could you provide analysis showing:

    * Model attention/activation patterns across different seasons
    * Performance comparison between different types of forests that show different seasonal patterns
    * Ablation study isolating seasonal pattern benefits from general data volume benefits

2. Geolocation Shift Benefits: Can you provide direct evidence that the model leverages Sentinel-2 geolocation shifts? Specifically:

    * Comparative analysis with models that cannot use shift information
    * Examples showing improved edge detection or spatial detail
    * Quantification of improvement specifically attributable to shift utilization

3. Growth Detection Validation: How reliable is the model at detecting forest growth? Please provide:

    * Validation using known growth areas
    * Comparison with ground measurements over time
    * Analysis of minimum detectable growth rate

**Relation To Broader Scientific Literature:**

The paper's contributions can be placed within the context of the broader scientific literature on Forest Monitoring and Remote Sensing:

* Builds upon established work using satellite data for forest monitoring
* Advances previous single-year height mapping efforts (Lang et al., Liu et al.) by incorporating temporal imagery.

**Theoretical Claims:**

Not Applicable

---

> ### Author Rebuttal · Authors · 2025-04-01
>
> Thank you for your detailed review. In our manuscript, we proposed seasonal variation and geolocation shifts as possible factors influencing the model's performance. We agree that some of our statements have been to explicit from a environmental perspective (e.g., to make use of the geolocation shifts given in Sentinel-2 time series data to improve model performance or the use of time series data to capture seasonal patterns). We regret these unprecise statements. We have revised the manuscript to present these as hypotheses rather than definitive claims. Below, we include further experiments and analyses that explore these factors in more detail. We hope that our comments and additional experiments address your concerns.
>
> ### Seasonal Pattern Analysis: How does the model specifically utilize seasonal information? Could you provide analysis showing:
> >
> > a) Model attention/activation patterns across different seasons
>
> The figure in https://ibb.co/1f8zh2gR shows activation patterns across months for different patches using Guided Attention [1]. We observe varying activation strengths across months and patches, suggesting the model processes temporal information differently by location. However, further research would be needed to confirm these hypotheses.
>
> [1] Striving for simplicity: The all convolutional net.
>
> > b) Performance comparison between different types of forests that show different seasonal patterns
>
> We evaluated our model separately on broadleaf and coniferous forests using the Copernicus Land Monitoring Service Forest Type Map (2018). These forest types show different seasonal patterns - broadleaf forests have distinct leaf-on/off periods while coniferous forests maintain constant canopy. The metrics below show our findings:
>
> |Model|Broadleaf MAE (m)|Coniferous MAE (m)|
> |-|-|-|
> |Lang et al.|5.44|5.11|
> |Liu et al.|7.01|6.91|
> |Pauls et al.|5.30|4.85|
> |Tolan et al.|10.43|11.73|
> |Turubanova et al.|8.60|8.43|
> |**Ours**|**4.57**|**4.11**|
>
>
>
> > c) Ablation study isolating seasonal pattern benefits from general data volume benefits
>
> Note that the number of training labels remains identical for all variants. To investigate the benefits of seasonal information independently from data volume, we conducted an ablation study comparing three models trained on different 4-month subsets:
>
> - Winter (Nov-Feb)
> - Summer (Jun-Sep)
> - Mixed (Jan-Feb, Aug-Sep)
>
> The results below show that using a mix of winter and summer months yields better validation performance.
>
> |Model Variant|Huber Loss (m)|
> |-|-|
> |Winter (Nov-Feb)|1.169 ± 0.003|
> |Summer (Jun-Sep)|1.13 ± 0.002|
> |Mixed (Jan-Feb, Aug-Sep)|1.122 ± 0.002|
>
> Thank you for this comment. We will provide these additional findings in the updated version of our manuscript.
>
>
> ### Geolocation Shift Benefits: Can you provide direct evidence that the model leverages Sentinel-2 geolocation shifts? Specifically:
> >
> > a) Comparative analysis with models that cannot use shift information
> > c) Quantification of improvement specifically attributable to shift utilization
>
> Direct analysis of geolocation shifts is challenging as they are inherently embedded in raw satellite data. However, our ablation study (Table 3) shows superior performance using raw vs composite data. Note that although the amount of input data changes, we keep the number of training labels constant.
>
> > b) Examples showing improved edge detection or spatial detail
>
> We provide a comparative figure (https://ibb.co/WpHqStqM) demonstrating enhanced edge detection across three model variants: 2D-Composite, 2D-Stack, and 3D-Stack. Additional examples of improved edge detection can be found in Figures 1, 4, and 8, as well as in our interactive Google Earth Engine application.
>
>
> ### Growth Detection Validation: How reliable is the model at detecting forest growth? Please provide:
> >
> > a) Validation using known growth areas
>
> While our model's ability to detect forest growth is constrained by the 4-year observation period, we observe clear growth signals in forest plantation regions like the Le Landes forest plantation in France (search of "Garein" in our GEE app).
>
> > b) Comparison with ground measurements over time
>
> We collected LiDAR data from the Vosges forest in France for 2020 and 2022 and analysed the measured vs predicted growth (https://ibb.co/YTbkXfpf). Our model is able to predict the growth of the trees, however the model uncertainty is high, which leads to "heavier tails" than what was measured. We have added these results to our manuscript.
>
> > c) Analysis of minimum detectable growth rate
>
> Given our model's MAE of 4.76m and GEDI's uncertainty of 0.98m, we estimate that reliable growth detection requires changes of at least 5m over the observation period, varying by region and forest type.
>
> We have also revised the paper about computational requirements, model performance and robustness across different forest types. Please let us know if you need any additional clarification.

---

### Official Review · Reviewer_sUWg · 2025-03-14

**Overall Recommendation:** 4

**Summary:**

The article describes a method for calculating canopy height using satellite data and reference values from GEDI LiDAR. The authors propose the use of a UNet network for regression. Using multispectral Sentinel images, they provide canopy height estimates for Europe between 2019 and 2022. Their R² is 0.819. When they only consider labels exceeding 7m, their R² is 0.591.

**Claims And Evidence:**

The authors provide an extensive discussion of their procedure to obtain their results. This discussion includes different options for constructing the training set, the use of various error measures, comparisons with other approaches, and the distribution of errors for different tree sizes. They also conduct a qualitative evaluation by presenting several examples.
However, some observations I have include:
*** How do you know that you are looking at trees (as opposed to buildings, for instance)?

**Essential References Not Discussed:**

A search yields the following articles related to the problem that are not mentioned:

1. **Satellite Image and Tree Canopy Height Analysis Using Machine Learning on Google Earth Engine with Carbon Stock Estimation**

**Experimental Designs Or Analyses:**

The article does not provide weights or data to verify the claims, although the authors have stated that they will do so once the paper is accepted. The included code appears to be reasonable.

**Methods And Evaluation Criteria:**

The authors construct their database, which consists of Sentinel-1 and Sentinel-2 images as predictors and GEDI LiDAR as the reference value (why is Sentinel-1 not mentioned in the abstract?). They then train their UNet using different error measures (*** Please explain why MSE≠RMSE\sqrt{\text{MSE}} \neq \text{RMSE}?).
Even though the authors do not provide their weights or datasets (they state that they will do so upon acceptance of their publication), the code provided appears reasonable.

**Other Comments Or Suggestions:**

***what is the normalizing divisor in Table 1? Where does it come from?


*** provide R2 in Table 2. (no need to include both MSE and RMSE)



*** I would recommend training, validating and testing in non-overlapping different years



*** seperate, Appendix A

**Other Strengths And Weaknesses:**

This article presents an important application to the study of forests. The authors scaled their solution to cover Europe, and their results appeared to advance the state of the art.
However, I am uncertain about how their study specifically returns to tree height rather than the height of objects in general, such as buildings.
The authors do not provide weights or data, although they state that they will make it available upon acceptance of the paper. They provide code that seems to be well structured. The details provided would allow for replicating their approach.

**Questions For Authors:**

1. How does the model distinguish between trees and other tall objects (e.g., buildings)?



2. What are the sources of error when estimating tall trees?


3. Why not use stratified evaluation based on different forest types?



4. What post-processing steps were used to ensure temporal consistency?

**Relation To Broader Scientific Literature:**

The paper
1. introduces a multi-temporal approach to canopy height estimation. Other studies have used
 optical data (Schwartz et al., 2024)
2. utilices Sentinel-2 time series instead of median composites( Pauls et al., 2024)
3. develops a 3D U-Net model that improves performance over prior 2D approaches

4. achieving state-of-the-art results in canopy height prediction at 10m resolution.
(Liu et al., 2023; Pauls et al., 2024)
5. provides publicly available code, but not weigths nor data.

**Theoretical Claims:**

The article does not include a relevant theoretical proof or one that needs to be tested. However,  I would recommend training, validating, and testing in different years, in non-overlapping.

---

> ### Author Rebuttal · Authors · 2025-04-01
>
> Thank you for your positive and thorough review. Let us address your concerns and questions in detail.
>
> > How do you know that you are looking at trees (as opposed to buildings, for instance)?
>
> That is a very good and true observation, indeed, given that GEDI measures the height of all objects, we cannot tell whether we are estimating a tree or a building: buildings are also measured at some height, therefore in cities one can see height prediction even in the absence of trees. This however is common practice in the remote sensing community. Canopy height maps are mainly used for forest monitoring and carbon stock estimation and both applications apply a forest mask before further use, which could be itself generated by a segmentation model; which is however a different problem we do not address here. While this is common practice, we will make this more clear in the paper to avoid confusion.
>
> > why is Sentinel-1 not mentioned in the abstract?
>
> That was not intentional, thank you for pointing it out. We have updated our manuscript accordingly.
>
>
> > Please explain why sqrt{\text{MSE}} \neq \text{RMSE}?
>
> We individually calculate the MSE and RMSE for each of the 1,500 validation patches and average them afterwards (weighted average based on the number of labels in each patch). This is why there can be a difference. Thank you for this remark. We will provide additional explainations in our manuscript.
>
>
> > Even though the authors do not provide their weights or datasets (they state that they will do so upon acceptance of their publication), the code provided appears reasonable.
>
> Please note that the predictions can be viewed and downloaded from the Google Earth Engine (GEE) website linked in the paper. We are happy to share all the code to reproduce our results upon acceptance. Sharing terabytes of data in an anonymous way right now is challenging.
>
> > I would recommend training, validating, and testing in different years, in non-overlapping.
>
> We tested this approach under the "-2020" setup, where we train only on 2020 data and evaluate on data from 2019, 2021 and 2022 (cf. e.g. Table 3 in the paper). We have made this more explicit in our manuscript, thank you.
>
>
> > A search yields the following articles related to the problem that are not mentioned: Satellite Image and Tree Canopy Height Analysis Using Machine Learning on Google Earth Engine with Carbon Stock Estimation
>
> Thank you very much. We have added the reference to the background section of our manuscript.
>
>
> > what is the normalizing divisor in Table 1? Where does it come from?
>
> We found it beneficial to rescale the data to a range that is more suitable for the model (i.e., we divide the input data by this divisor). To that end, we manually inspected the data to find out value ranges that contain valuable data, in particular because standard normalization did not work due to problem with atmospheric distortions and cloud cover.
>
> > provide R2 in Table 2. (no need to include both MSE and RMSE)
>
> Thank you for the suggestion, we have added R2 to Table 2.
>
>
> > seperate, Appendix A
>
> Fixed, thank you!
>
>
> > What are the sources of error when estimating tall trees?
>
> Although we do not know the exact reason, we suspect that it has to do with the following two reasons:
> 1. Due to the natural distribution of trees in Europe, tall trees are less common, creating a skew in the label distribution.
> 2. Tall trees naturally have a higher canopy density (e.g. more leaves, branches, etc.), which leads to a higher fraction of LIDAR photons/measurements not penetrating the canopy. In that case, the photons do not reach the ground and we do not have a usable measure of the tree height (this filtering is already applied by GEDI, not by us).
>
>
> > Why not use stratified evaluation based on different forest types?
>
> Thank you for this suggestion. While our paper focused on overall performance metrics, we have now included a detailed analysis comparing broadleaf and coniferous forests (see our response to reviewer qZ94). We welcome your input on additional forest categories that would be valuable to evaluate.
>
>
> > What post-processing steps were used to ensure temporal consistency?
>
> We use a quadratic-spline approach to smooth the predictions over time, but only for visualization purposes.
>
> We hope to have addressed all your remarks. Thank you again. If you have any further questions or concerns, please let us know.

---

### Official Review · Reviewer_1141 · 2025-03-14

**Overall Recommendation:** 4

**Summary:**

This paper introduces a novel deep learning approach for generating high-resolution, large-scale temporal canopy height maps across Europe using satellite imagery, specifically leveraging Sentinel-2 time series data and GEDI LiDAR measurements as training data. The proposed method significantly improves accuracy and resolution, delivering consistent 10-meter spatial resolution canopy height predictions from 2019 to 2022, which allows for the tracking of temporal dynamics such as deforestation and forest growth. By using a 3D U-Net model architecture with monthly temporal stacks, the model effectively captures seasonal variations and geolocation shifts, demonstrating superior performance compared to existing approaches, especially in accurately estimating the height of tall trees critical for carbon stock assessment and ecological analysis. In general, this paper makes a great contribution in the training dataset and canopy height data products.

## Update after Rebuttal
The authors' response to my questions fully clarified my concerns. This is a good paper that utilizes well-established methods in an application area that resolves practical questions in sustainability. Thus, this paper is a good fit for ICML's Application-Driven Machine Learning track and deserves acceptance.

**Claims And Evidence:**

The authors present three central claims, all of which are supported by quantitative and qualitative results:
1. **A model capable of tracking forest height changes**: The paper features the first 10 m resolution temporal canopy height map of the European continent for 2019–2022, as shown in the Earth Engine app. Thus, comparing those annual maps can be used to track changes in forest height. In addition, Figure 6 provides additional qualitative examples of change tracking.
2. **More accurate measurements and finer spatial details than previous studies**: The performance of the model against baselines is comprehensively evaluated through the experiments in 4.3.
3. **12-month timeseries is helpful than a single composite**: Table 2 and Table 3 substantiate this claim.

**Essential References Not Discussed:**

More references to remote sensing timeseries understanding literature such as [1] can be discussed.

[1] Tarasiou, M., Chavez, E. and Zafeiriou, S., 2023. Vits for sits: Vision transformers for satellite image time series. In Proceedings of the IEEE/CVF Conference on Computer Vision and Pattern Recognition (pp. 10418-10428).

**Experimental Designs Or Analyses:**

I reviewed the experiment design, especially the validation dataset. My only concern is the possible spatial autocorrelation in the validation dataset since the validation dataset is generated by randomly selecting tiles. Is it possible that two very close tiles share similar canopy heights?

**Methods And Evaluation Criteria:**

The evaluation protocol of this paper conforms with prior works in forest canopy height estimation. The proposed method (3D Unet for mapping tree height from satellite timeseries) also makes sense conceptually.

**Other Comments Or Suggestions:**

Although this paper mainly focuses on methodology and dataset, I encourage the author to include scientific conclusions about forest health or deforestation of the European continent, if any, from analyzing the trend of yearly forecast canopy height maps generated by the model.

**Other Strengths And Weaknesses:**

This paper is well written and has great contributions in its training dataset and the final data product produced. Please see other comments and questions for the weaknesses.

**Questions For Authors:**

1. Line 152: What’s the reason for including coastal aerosol (B01) and water vapour (B09) bands?
2. Line 185: Could the authors clarify the sparsity of the labels? What percentage of the pixels have a canopy height label derived from GEDI?
3. Line 197: How are 2.56 km × 2.56 km patches created? Are they created by buffering GEDI point measurements?
4. Line 199: Do “month images” refer to month composites or one image selected within a month? If the latter, could the authors clarify the reasons for not using monthly composites?
5. Line 207: Is smoothing applied before calculating validation metrics or just for the final mapping visualization?
6. Line 240: Have the authors considered possible spatial autocorrelations in the validation dataset? For example, two randomly selected points can have similar canopy heights.
7. Have the authors considered other architectures for remote sensing timeseries such as [1]?

[1] Tarasiou, M., Chavez, E. and Zafeiriou, S., 2023. Vits for sits: Vision transformers for satellite image time series. In Proceedings of the IEEE/CVF Conference on Computer Vision and Pattern Recognition (pp. 10418-10428).

**Relation To Broader Scientific Literature:**

The paper advances the field of tree canopy height estimation by introducing a novel deep learning model capable of producing temporal canopy height maps at 10 m resolution over large spatial scales. Unlike prior studies that mainly focused on single-year canopy height predictions, this work addresses the critical gap of modeling temporal dynamics, crucial for tracking ecological changes and carbon stocks. It builds upon previous research by using Sentinel-2 monthly image stacks instead of aggregated median composites, thereby leveraging seasonal vegetation patterns and subtle geolocation shifts to enhance accuracy. Furthermore, this approach significantly surpasses existing methodologies, such as those of Liu et al. (2023) and Tolan et al. (2024), especially in accurately estimating taller trees, which are essential for precise biomass estimations.

**Theoretical Claims:**

N/A

---

> ### Author Rebuttal · Authors · 2025-04-01
>
> Thank you for your thorough review and for acknowledging the contributions of our work. Let us address your concerns and questions one by one.
>
> > More references to remote sensing timeseries understanding literature such as [1] can be discussed.
>
> Thank you for the suggestion. We have added additional references to our manuscript.
>
> > Although this paper mainly focuses on methodology and dataset [...]
>
> We agree this is important. While our main focus is providing an openly available model for others to analyze, we did conduct some initial analysis: By tracking pixels where height decreased from >8m to <5m between years (indicating potential deforestation), we found affected areas of 9,747.9 km² (2019-2020), 7,729.1 km² (2020-2021), and 15,942.5 km² (2021-2022). We have added these findings to our manuscript.
>
> > Line 152: What’s the reason for including coastal aerosol (B01) and water vapour (B09) bands?
>
> We decided to include all L2A (bottom-of-the-atmospohere) information, which includes all bands except for B10. We agree that B01 and B09 might not necessarily be relevant for the task at hand. To analyze their impact, we ran additional experiments to assess their importance. The following table reports the results on the validation part of our dataset (L1>15m refering to the L1 loss for all labels that exceed 15m):
>
> | Configuration         | L1 (m)         | L1>15m (m)     | L1>20m (m)     | L1>25m (m)     | L1>30m (m)     | L2 (m)         |
> |-----------------------|----------------|----------------|----------------|----------------|----------------|----------------|
> | Without B01 and B09   | 1.991 ± 0.002 | 4.837 ± 0.008 | 5.476 ± 0.010 | 7.384 ± 0.008 | 11.406 ± 0.004 | 22.281 ± 0.037 |
> | Including B01 and B09 | 1.992 ± 0.003 | 4.830 ± 0.014 | 5.460 ± 0.015 | 7.364 ± 0.029 | 11.384 ± 0.031 | 22.277 ± 0.053 |
>
> As it can be seen, B01 and B09 only have little impact on the results and are, hence, candidates to be removed from the set of input channels. We will discuss these findings in the updated version of our manuscript.
>
> > Line 185: Could the authors clarify the sparsity of the labels? What percentage of the pixels have a canopy height label derived from GEDI?
>
> GEDI measures roughly 4% of Earth's surface. For our 256x256 pixel training samples, only about 100 pixels (0.15%) have usable GEDI labels from the same year, due to noise and the need to match measurements temporally with satellite imagery.
>
> > Line 197: How are 2.56 km × 2.56 km patches created? Are they created by buffering GEDI point measurements?
>
> We first make sure to only take image patches from our training areas to not have an overlap with the validation patches. We then randomly select a 2.56km x 2.56km area and load all GEDI measurements within that area. Since we only have the coordinates of the GEDI measurements, we assign them to the closest Sentinel-pixel. We hope that this clarifies your questions. Please let us know if that is not the case.
>
> > Line 199: Do “month images” refer to month composites or one image selected within a month? If the latter, could the authors clarify the reasons for not using monthly composites?
>
> When creating the 12-months image stack, we select one of the images within each month, namely the one with the least amount of cloud cover. We decided not to use monthly composites, following Wolters et al., who showed that not using composites allows the model to learn finer details, possibly due to small geolocation shifts in the satellite images.
>
> > Line 207: Is smoothing applied before calculating validation metrics or just for the final mapping visualization?
>
> Smoothing is only applied for the visualizations.
>
> > My only concern is the possible spatial autocorrelation in the validation dataset [...]
>
> Thank you for raising this point. Indeed, there are spatial correlations between the different spatial areas. Note, however, that the learning scenario can be seen as a transductive learning setting, where one already has access to the (test) input images (but not the ground-truth labels). From that perspective, it is even valid to include the geolocation of an image stack as input. Note that the overall goal is to fill the remaining "gaps" that are not covered by GEDI groundtruth labels. We hope that this answers your question. Please let us know if there are still remaining concerns from your side.
>
> > Have the authors considered other architectures for remote sensing timeseries such as [1]?
>
> We have considered other segmentation architectures and backbones, but we found that the 3D extension of the U-Net architecture to work surprisingly well. U-Nets are efficient and in our setting, yield good results. We will consider exploring more complex architectures in future work.
>
> We hope to have clarified all concerns and questions, please let us know if further clarification is needed. Thanks!

---

> > ### Comment · Reviewer_1141 · 2025-04-04
> >
> > I thank the authors for their detailed response. My concerns have been addressed, and I have raised my scores accordingly. I think this is an excellent application paper and deserves acceptance.

---

### Decision · Program_Chairs · 2025-05-01

**Decision:**

Accept (poster)

**Comment:**

This paper received divergent scores after the rebuttal. Reviewers 1141, sUWg, and edjr provided positive recommendations, whereas reviewer qZ94 maintained a weak reject rating.

Reviewers 1141 and sUWg praised the paper for its significance in generating high-resolution, large-scale temporal canopy height maps across Europe. They highlighted its innovative integration of 12-month Sentinel-2 time series data with GEDI LiDAR measurements to produce consistent 10 m resolution maps, which are properly validated both quantitatively and qualitatively. Reviewer edjr further praised the paper’s valuable contributions to time-series forest monitoring and its practical relevance for ecological analysis and carbon stock assessment. In contrast, reviewer qZ94 raised concerns regarding technical limitations and the insufficient depth of analysis supporting some of the claims. Although the authors’ rebuttal addressed several of these issues, reviewer qZ94 remained concerned about the reliance on existing methodologies.

After carefully reviewing the paper, reviews, and rebuttal, the Area Chair acknowledges that reviewer qZ94’s concern is valid. However, the extensive experimental results, comprehensive comparisons, and a well-articulated rebuttal demonstrate that the work possesses substantial technical significance and real-world impact. Based on these considerations, the Area Chair recommends acceptance, with the expectation that the authors will address the reviewers' comments in the final version of the paper.